# *Bacillus halophilus* BH-8 Combined with Coal Gangue as a Composite Microbial Agent for the Rehabilitation of Saline-Alkali Land

**DOI:** 10.3390/microorganisms13030532

**Published:** 2025-02-27

**Authors:** Weilin Bi, Yixuan Sun, Zhipeng Yao, Zhe Zhao, Yusheng Niu

**Affiliations:** 1Institute of Biomedical Engineering, College of Life Sciences, Qingdao University, Qingdao 266071, China; bwl04406@163.com (W.B.); yisuken@sina.com (Y.S.); 2Binzhou Academy of Agricultural Sciences, Binzhou 256600, China; yzp963@126.com; 3Research Institute of Modern Agricultural Industry Innovation in Yellow River Delta Saline-Alkali Land, Dongying Vocational College, Dongying 257000, China

**Keywords:** *Bacillus halophilus*, coal gangue, saline-alkali land, composite microbial agent, microbial community

## Abstract

Saline-alkali land represents a crucial reserve of arable land essential for ensuring food security. However, there remains a significant deficiency in converting saline-alkali land into productive cultivated or grazing areas. Microbial agents hold substantial potential for the reclamation of saline-alkali soils. In this study, a moderately halophilic bacterium, *Bacillus halophilus* BH-8, was screened from coastal saline soil. We combined strain BH-8 with coal gangue to create a composite microbial agent, which was shown to effectively increase the levels of available nitrogen, available phosphorus, available potassium, and organic matter, while reducing the pH value of saline-alkali soils. Moreover, it significantly enhanced the activity of various enzymes and altered the microbial community composition in the soil, notably increasing the abundance of *Pseudomonas* and *Bacteroidota*. These results demonstrate the application value of this composite microbial agent for rehabilitating saline-alkali land and highlight the potential of the BH-8 strain as a promising candidate for microbial agent research.

## 1. Introduction

Over-salinization of the soil can destroy the physical and chemical structure of the soil, causing significant impacts on the growth and yield of crops [1,2]. In areas with soil salinization, the soil shows characteristics of high salt content and strong alkalinization. Soil salinization will also reduce the permeability of the soil, affect the water absorption activity of plants, and lead to physiological dehydration and other problems [3,4]. Saline-alkali stress in salt-affected soils leads to an overaccumulation of reactive oxygen species in plants, causing cellular tissue damage and hindering normal plant growth [5]. It also has a negative impact on soil microbial communities, deteriorating soil biological properties and impeding the growth and development of plants and crops [6,7,8]. In recent years, the phenomenon of soil salinization has been exacerbated by natural and human factors. Currently, the area of saline-alkali land worldwide exceeds 833 million hectares, accounting for 8.7% of the Earth’s surface, and the area of soil salinization is increasing at a rate of over 2 million hectares per year [9]. As a country with widespread saline-alkali land distribution, the area of saline-alkali land in China is equivalent to 40% of the total arable land in the country [10], which has severely impacted food and environmental security. Therefore, the issue of managing saline-alkali land urgently needs research and solutions.

In recent decades, previous researchers have carried out a lot of studies on the management and rehabilitation of saline soils, mainly centered on physical improvement, chemical improvement, and biological improvement [11,12]. Physical improvement is mainly through deep plowing [13], straw mulching [14], and other traditional methods to improve the soil structure to enhance soil permeability, reduce evaporation, and improve the efficiency of soil salinity leaching. Heng et al. [15] showed that the combination of underground pipelines and vertical well drainage measures can improve the agrosoil ecosystems of arid desert areas and increase the utilization value of saline-alkaline land. And long-term sand mulching can significantly improve the physicochemical properties of soil in the tillage layer of saline cultivated land [16]. Chemical improvement refers to the addition of chemical amendments such as biochar [17], marl [18], and desulfurization gypsum [19] to saline-alkaline soils, which can be replaced with sodium and chloride ions in the soil, and can also improve soil porosity, inhibit the aggregation of salts to the topsoil, and improve the physicochemical properties of the soil and the structure of the microbial community. For example, Yang et al. [20] found that organic acid amendments, such as He Kang (HK), in areas such as northern Shaanxi Province, had significant effects on the improvement of secondary saline soils. Biological amendment refers to the management and improvement of saline soils through the application of organic fertilizers [21], microbial fertilizers [22,23,24], and planting salt-tolerant plants [25,26,27]. The gradual and mature application of biological improvement technologies offers new approaches for the rehabilitation of saline and alkaline lands. For example, microbial fertilizers such as *Priestia aryabhatai* JL-5 [28] have shown potential in saline soil improvement. In addition, the application of some salt-tolerant microorganisms can effectively exclude Na^+^ ions from the rhizosphere by increasing the absorption of K^+^, alleviating salt stress, and promoting the successful cultivation of plants [29].

Coal gangue consists of waste tailings produced during the coal mining process, accounting for 15–20% of coal production [30]. However, the utilization of coal gangue is insufficient, making it one of the largest industrial solid wastes [31]. The accumulation of coal gangue not only occupies a large amount of land resources but also causes many severe problems, such as soil degradation, spontaneous combustion, the release of toxic gases, heavy metal pollution, and geological disasters [32,33]. At the same time, due to reasons such as soil salinization, serious soil degradation has occurred in many regions of the world [34], and soil conditioners are needed to supplement the soil. The use of commercial soil conditioners will generate a large cost [35], and previous studies have shown that modified coal gangue is expected to effectively improve soil quality and show great potential in reducing carbon emissions and promoting soil carbon sequestration [36]. At the same time, due to its high porosity and strong water absorption, coal gangue can change the water content and stability of the soil aggregate structure, and research on its application in improving saline-alkali soils has been ongoing in recent years. In addition, coal gangue is rich in organic matter and inorganic elements such as nitrogen, phosphorus, and potassium, but its organic matter is highly stable, and the decomposition and transformation rate of inorganic elements is low, leading to poor improvement effects on saline-alkali soils. Microorganisms can decompose and transform minerals, continuously release mineral ions, and effectively improve the decomposition and transformation rate of mineral elements in coal gangue. *Halophiles* are a group of microorganisms that can only grow in saline-alkali environments, divided into moderately halophilic bacteria and extreme halophilic bacteria according to the required salt content. Studies have shown that *Pseudomonas*, as a *halophile*, can significantly increase the content of available nitrogen and phosphorus in saline-alkali soils [37], but the application of *halophiles* in the decomposition and transformation of inorganic elements in coal gangue has not been studied.

In this study, to explore the effect of the combined application of coal gangue and halophilic bacteria in saline-alkali soil remediation, a moderately halophilic bacterium, *Bacillus halophilus* BH-8, was screened and isolated from coastal saline soil. The BH-8 strain was inoculated onto coal gangue to create a composite microbial agent for ameliorating saline-alkali soils. Subsequently, the fundamental physical and chemical properties of the soil were analyzed, including available nitrogen (AN), available phosphorus (AP), available potassium (AK), the content of organic matter (OM), the pH value, the activities of relevant enzymes, and the composition of the microbial community within the soil ecosystem. Additionally, we investigated the impact of applying the composite microbial agent on gene functions of soil microbial communities. The above analyses were used to confirm whether this novel composite microbial agent plays a role in rehabilitating saline-alkali land.

## 2. Materials and Methods

### 2.1. Site Description, Sampling, and Soil Characterization

The coastal saline soil was collected from Dongying City, Shandong Province. Total salt, density, conductivity, and total nitrogen content in the soil were tested by the Hangzhou Yanqu Information Technology Co., Ltd. A random sampling method was adopted to sample soil from 0–20 cm, mix the soil sample evenly, air dry it naturally, and then pass it through a 2 mm sieve to remove plant debris, stones, and other impurities from the soil sample.

The experiment was carried out indoors. A total of 200 g air-dried soil samples were put into a black plastic flowerpot, and the bottom of the flowerpot was lined with filter paper to prevent the loss of soil samples. We set up a control group (without application of soil amendment), a group treated only with coal gangue, a group treated only with the BH-8 microbial agent, and a group treated with a composite microbial agent. The same amount of sterile water was added every 3 days to maintain 40% water capacity for 21 days, with three replicates for each treatment, respectively.

### 2.2. Isolation of Halophilic Bacteria

The enriched culture medium in this experiment is an optimized LB medium, containing 5 g of yeast extract, 10 g of peptone, 5 g of magnesium sulfate heptahydrate, 2.5 g of potassium chloride, 0.05 g of ferric sulfate heptahydrate, and 3 g of sodium chloride per liter of medium. One g of soil sample was added to a 250 mL conical flask containing 100 mL enriched culture medium, mixed, and shaken at 30 ℃ for 2–3 days. We used the gradient diluted bacterial suspension and spread it on plates, and incubated it at 30 ℃. We selected well-grown colonies with a loop and streaked them on solid culture media using the cross-streaking method to obtain pure single colonies. NaCl concentrations in the culture medium were gradually increased to screen for halophilic bacterial strains. Colony morphology of the halophilic bacterial strains was recorded, and their physiological and biochemical characteristics were analyzed. Specifically, in this study, the NaCl concentrations employed were 3 g/L, 5 g/L, 8 g/L, 10 g/L, 12 g/L, 15 g/L, 18 g/L, and 20 g/L.

### 2.3. 16S rDNA Gene Sequencing Analysis

The total DNA of strain BH-8 was extracted using a bacterial genomic DNA kit (TIANGEN BIOTECH Co., Ltd., Beijing, China), and the extracted DNA group product was subjected to agarose gel electrophoresis, which showed clear bands, proving that the extracted bacterial DNA group was intact and unbroken. The extracted DNA products were sent out for 16S rDNA sequencing (Shanghai Sangon Biotech Co., Ltd., Shanghai, China). The 16S rDNA sequencing results were compared with known species sequences in Genbank, and a phylogenetic tree of 16S rDNA analysis was constructed using Mega 11 software. The sequence was submitted to Genbank with the accession number BH-8 PP657625.

### 2.4. Preparation of Composite Microbial Agent

The coal gangue was sourced from Henan, China. The chemical composition of the coal gangue was determined using X-ray fluorescence (XRF) analysis. After removing impurities, the coal gangue was sieved to obtain particles of 0.15 mm, 0.3 mm, and 1 mm in size. The BH-8 strain was cultured with the optimized LB medium; following centrifugation, the bacteria were resuspended in sterile water to a concentration of 10^9^ CFU/mL. The resuspended culture was inoculated into LB medium and co-cultivated with coal gangue of different concentrations (0 wt%, 5 wt%, 10 wt%, 15 wt%, 20 wt%, 25 wt%) and particle sizes (0.15 mm, 0.3 mm, 1 mm) at 30 °C. The wt% refers to the weight percentage of coal gangue in the culture.

### 2.5. Observation with Microscope

*Bacillus halophilus* BH-8 spore observation was performed using the Schaeffer–Fulton staining method. The sample was placed on a slide and stained sequentially with malachite green (which requires heating), rinsed, counterstained with safranin, rinsed again, dried, and then observed under a light microscope. The spores appeared blue-green, while the vegetative cell bodies appeared red.

During the co-culture process of Bacillus halophilus BH-8 with coal gangue, the bacterial bodies and coal gangue were enriched by centrifugation, washed with PBS solution three times, then fixed with glutaraldehyde solution at 4 °C for 12–14 h. After this, dehydration was performed using 30%, 50%, 70%, 90%, and 100% ethanol, each for 10 minutes. Conductive coating was then imaged using a VEGA3 scanning electron microscope (SEM) (TESCAN China Co., Ltd., Shanghai, China).

### 2.6. Analysis of Soil Physical and Chemical Properties

The pH of air-dried soil samples was measured using a pH meter, with a water-to-soil ratio of 2.5:1 (weight/volume). The AN content in the soil was measured using the alkaline-diffusion method after initial titration with 1 mol/L NaOH and 0.1 mol/L H_2_SO_4_. The AK was extracted using a mixed acid solution of 0.05 mol/L HCl and 0.0125 mol/L H_2_SO_4_ mixed acid, and the K^+^ concentration in the extract was determined by ion chromatograph. The AP content was determined using an ultraviolet spectrophotometer. The OM content was determined using the potassium dichromate volumetric method.

### 2.7. Analysis of Soil Enzyme Activity

Soil catalase (CAT) activity was determined by potassium permanganate titration. We weighed 2 g of air-dried soil sample into a conical flask. We added 40 mL of distilled water and 5 mL of 0.3% H_2_O_2_ solution, and shook it for 20 min. We then added 10 mM H_2_SO_4_ to terminate the reaction immediately, filtered and took 25 mL of filtrate, add 10 mM of H_2_SO_4_ to acidify, and titrate with KMnO_4_ to a pale pink endpoint. Activities of soil urease (URE), acid phosphatase (ACP), nitrate reductase (NR), and alkaline phosphatase (ALP) were determined using the corresponding kit (Shanghai Bolsen Biotechnology Co., Ltd., Shanghai, China). Briefly, urease activity was quantified via ammonia release using indophenol blue formation (578 nm). Acid and alkaline phosphatase activities were measured using p-nitrophenyl phosphate hydrolysis (405 nm), with buffers adjusted to pH 5.0 and 9.5, respectively. The nitrate reductase activity was assessed by nitrite production via sulfanilamide and N-(1-naphthyl) ethylenediamine dihydrochloride (NED) colorimetry (540 nm).

### 2.8. Metagenomic Sequencing Analysis

Genomic DNA was extracted using a DNA Extraction Kit (TIANGEN BIOTECH Co., Ltd., Beijing, China) and delivered to the company (Beijing Novogene Co., Ltd., Beijing, China) for sequencing after library construction with Illumina PE150. Clean data were assembled and analyzed using MEGAHIT (v1.0.4) software. ORF prediction for each sample was performed using MetaGeneMark to obtain basic information statistics, core-pan gene analysis, inter-sample correlation analysis, and Wayne’s plot analysis of gene numbers. The relationship between samples and species was visualized using Circos.

### 2.9. Statistical Analyses

Bacterial growth was monitored by measuring the optical density at 600 nm (OD_600_). Growth curves were fitted using GraphPad Prism 9 software. SPSS version 26.0 was used to perform one-way ANOVA of enzyme activities, soil physical and chemical properties. An independent sample *t*-test (*p* < 0.05) was used for statistical comparisons, and GraphPad Prism was employed for data visualization. A heatmap representing the correlation between heat and species abundance was created as a stacked graph using ChiPlot (https://www.chiplot.online/ (accessed on 27 November 2023)). The soil microbial community structure was displayed using principal component analysis (PCA) and non-metric multidimensional scaling (NMDS). Redundancy analysis (RDA) was used to evaluate the relationships between the soil and microbial community. Soil microbial diversity was assessed using α-diversity. Soil microbial functions were annotated using the CAZy database, and related differential gene functional modules were screened using linear discriminant analysis (LDA).

## 3. Result

### 3.1. Screening and Identification of Strains

The soil electrical conductivity, density, total salt content, and total nitrogen content of the coastal saline-alkali soil used in this study were 91.7 ms/m, 2.64 g/cm^3^, 2.6 g/kg and 0.89 g/kg, respectively. Firstly, we isolated and identified a moderately halophilic bacterium from the above soil, which we designated as BH-8. As shown in Figure 1a, the colonies of BH-8 exhibit a distinctive yellow pigmentation, and are non-flagellated and non-motile. The colonies present a moist and smooth surface texture, and Gram staining reveals a positive reaction. The physiological, biochemical, and enzymatic activity characteristics of this strain are shown in Appendix A. To identify the genus of strain BH-8, the 16S rDNA fragment was amplified and sequenced. Sequence analysis through BLAST (v2.14.0) against the NCBI database and construction of a phylogenetic tree revealed that strain BH-8 clusters with *Alkalihalobacillus hemicentroti* JSM076093 within the same branch. Additionally, strain BH-8 exhibits a high degree of homology with *Pseudalkalibacillus hwajinpoensis* SW-72, *Salibacterium nitratireducens strain* SMB4, and *Alkalihalobacillus algicola strain* AB423f (Figure 1b). In conclusion, strain BH-8 is identified as a member of the *Bacillaceae* family. Furthermore, BH-8 displayed remarkable salt tolerance as a moderately halophilic bacterium, withstanding at least 12% NaCl, and maintained a robust growth within the 2% to 5% NaCl range (Figure 1c). Microscopy analysis confirmed that BH-8 is a spore-forming bacterium (Appendix A). This characteristic probably enhances its survivability in harsh saline-alkali environments.

### 3.2. Co-Incubation of Coal Gangue and BH-8

XRF analysis indicates that the contents of SiO_2_, Al_2_O_3_, and Fe_2_O_3_ in our coal gangue are relatively high, and the main elements are O, Si, Fe, and Al (Appendix A). To explore the preparation conditions of the composite microbial agent, we examined the growth of BH-8 co-cultured with coal gangue at varied concentrations and particle sizes. The results showed that as the concentration of coal gangue increased, the growth of the strain was gradually inhibited (Appendix A). In addition, smaller grain sizes of coal gangue resulted in higher growth inhibition of BH-8 (Appendix A). The inhibition of BH-8 growth by smaller coal gangue particles can be attributed to their higher surface-area-to-volume ratio, which enhances the dissolution and diffusion of salts and potential heavy metals into the culture medium. This probably facilitates the influx of toxic elements into the cytoplasm and impairs metabolic processes critical for bacterial reproduction. Conversely, larger particles exhibit a lower exposure to the culture medium, thereby exerting a reduced toxic effect. SEM analysis further revealed that BH-8 cells preferentially adhere to larger gangue particles (1 mm) and form microcolonies (Appendix A), suggesting that larger gangue particles support microbial colonization. This may be because larger particles have deeper and larger pores that are favorable for bacterial adhesion. Finally, to maintain the normal growth of BH-8, we selected coal gangue with a particle size of 1 mm and a concentration of 5 wt% for preparing the composite microbial agent.

### 3.3. Improvement of Soil Physicochemical Properties and Enzymatic Activity

To investigate the effects of different soil amendments on the physicochemical properties of saline-alkali soil, we examined the levels of AN, AK, AP, and OM in soils. As shown in Figure 2a, the AN content of saline-alkali soil increased significantly when BH-8 and the composite microbial agent were applied compared to the control group. Specifically, on the first day after applying the composite microbial inoculant, the AN content in the soil was nearly four times that of the control group. Although the impact of microbial agents diminishes over time, the AN content treated with these agents for 21 days remained approximately twice as high as that in the control group. While the composite microbial agent outperformed the strain BH-8 alone, the latter still demonstrated a significant increase in the AN content in the soil. In contrast, the application of coal gangue alone did not show any significant effects.

Similarly, both the application of BH-8 alone and our composite microbial agent could effectively increase the levels of AK, OM, and AP in saline-alkali soil (Figure 2b,c and Appendix A). However, the benefits of using coal gangue alone for saline-alkali soil reclamation were not significant, with only a short-term increase observed in organic matter content (Figure 2c). Notably, the increase in OM content in the control group may be due to the growth and proliferation of microorganisms in the soil over the extended treatment period. In addition, to directly assess the improvement effects of microbial agents on alkaline soil, we measured the pH value of soils under different treatment conditions. The results indicated that both the application of BH-8 alone and our composite microbial agent significantly reduced the pH of alkaline soil to approximately 8.5 and maintained a relatively lower pH after 21 days of treatment (Figure 2d). These results indicate that the application of BH-8 could significantly improve the soil physicochemical properties, and suggest its potential for enhancing the nutrient content of coastal saline soil.

To analyze the effects of different treatment methods on soil enzyme activity, we measured the activity of soil catalase, urease, acid phosphatase, nitrate reductase, and alkaline phosphatase. The results show that the composite microbial agent effectively increased the activity of catalase, urease, acid phosphatase, and nitrate reductase in the soil within 2 to 3 weeks (Figure 3a–d). However, there was no significant change in the activity of alkaline phosphatase in the composite microbial agent group (Figure 3e). While the effect of BH-8 alone was not as dramatic as that of the composite microbial agent, it also effectively improved the activity of soil catalase, urease, and acid phosphatase when compared to the controls (Figure 3a–c). However, using coal gangue alone did not show significant effects (Figure 3). These results indicate that the combinate use of *Bacillus halophilus* BH-8 and coal gangue had a significant effect on improving soil enzyme activity and promoting soil health.

### 3.4. Alterations in Soil Microbial Community Structure and Diversity

The number of shared and unique genes among different treatment samples is shown in Figure 4a. Among these, 2,415,616 genes were shared across the four groups, accounting for 72.66% of the total number of genes. This indicates that the majority of genes are expressed under different treatment conditions. Notably, the BH-8 group and the composite microbial agent group exhibited a significant increase in the number of unique genes compared to the control group, while the coal gangue group showed only a slight increase. The analysis of gene abundance correlation among samples indicates that the BH-8 group and the composite microbial agent group had a lower correlation with the control group. In contrast, the coal gangue group shows a higher correlation with the control group (Figure 4b). These results suggest that the application of BH-8 probably activates the expression of certain specific genes, thereby influencing the metabolism of the microbial community.

The similarity of microbial communities in different soil samples was studied using principal component analysis (PCA) and non-metric multidimensional scaling (NMDS). As shown in Appendix A, the microbial community structure of the composite microbial agent group showed the lowest similarity to the control group, followed by the BH-8 group. This indicates that the application of the BH-8 strain had a significant reshaping effect on the soil microbial community structure. In contrast, the coal gangue group exhibited little difference compared to the control group, suggesting that the impact of coal gangue on the soil microbial community structure is relatively minor.

Microbial richness and diversity in different soil samples were calculated using α-diversity indices. As shown in Table 1, the α-diversity measurements showed that both the Shannon and Simpson indices of the BH-8 group and the composite microbial agent group were lower than those of the control group, while the Chao1 and ACE indices were higher. However, the α-diversity indices of the soil treated with coal gangue alone did not change significantly. On one hand, the Simpson index approached 1, whereas the Shannon index significantly decreased, indicating the application of strain BH-8 and coal gangue reduced the microbial diversity and evenness, with a potential increase in the dominance of certain species. On the other hand, the significant increase in the Chao1 estimate and ACE estimate indicated that the richness of the microbial community in the soil was greatly increased after the application of BH-8 and the composite bacteria of coal gangue. Additionally, the effect of the composite microbial agent group was more obvious than that of the BH-8 group alone, indicating that the combined use of BH-8 and coal gangue intensified the impact on the diversity of soil microbial communities.

### 3.5. Effects on Soil Microbial Community Composition

To study the impact of different treatments on the soil microbial community composition, we analyzed the microbial composition at the phylum, class, and genus levels. The relative species histogram based on phylum level reveals the top 10 phyla with the highest relative abundance (Figure 5a), *Pseudomonadota*, *Bacteroidota*, and *Actinomycetota* experienced significant changes in the phylum composition of the BH-8 group and the composite microbial agent group. Specifically, *Pseudomonadota* occupied a dominant position in all treatments and showed an increase of about 10.5% after the application of BH-8 and the composite microbial agent. Compared with the control, the relative abundance of *Bacteroidota* in the soil treated with BH-8 and the composite microbial agent increased by 6.86% and 15.35%, respectively, while the relative abundance of *Actinomycetota* decreased by 3.92% and 9.02%, respectively (Figure 5a). To study species with significant differences between groups, we drew a heat map of the abundance clustering of differential species between groups (Figure 5b). It can be observed from Figure 5b that the abundance of the *Bacteroidots* phylum after combined treatment shows the most significant change compared to the other groups. Simultaneously, *Pseudomonadota*, *Bacillota*, and *Uroviricota* also showed a significant increase.

As shown in Figure 5c, the histogram of different species abundance clustering based on class level revealed that the dominant class in all samples was *Gammaproteobacteria*, which showed a significant upward trend in the BH-8 group and the composite microbial agent group. In addition, the relative abundance of *Flavobacteriia*, *Cytophagia*, and *Bacilli* increased significantly. In contrast, the relative abundance of *Alphaproteobacteria*, *Actinomycetes*, and *Acidimicrobia* decreased significantly. The relative species abundances at order and family levels are shown in Appendix A. After the application of BH-8 and the composite microbial agent, the relative abundances of *Pseudomonadales*, *Flavobacteriales*, and *Chromatiales* were significantly increased at the order level; the relative abundances of *Flavobacteriaceae*, *Chromatiaceae*, and *Pseudomonadaceae* were significantly increased at the family level. At the genus level (Figure 5d), the relative abundances of *Salinimicrobium*, *Antarcticibacterium*, and *Arsukibacterium* were significantly increased, and the relative abundances of *Nocardioides* and *Gillisia* were significantly decreased after the treatment of BH-8 and the composite microbial agent. Overall, the microbial composition in soils treated only with coal gangue exhibited minimal changes, whereas the most significant alterations in microbial composition were observed in soils treated with the composite microbial agent. These findings suggest that applying BH-8 has a significant impact on the microbial community composition in saline-alkali soils, with the addition of coal gangue intensifying this impact. Notably, the application of our composite microbial preparation can lead to a significant increase in the proportion of certain microbial communities such as *Bacteroidota* and *Pseudomonadota*, which are beneficial for soil optimization and promoting plant stress resistance.

### 3.6. Correlation Between Soil Physicochemical Properties and Microbial Community Composition

To study the relationship between the physicochemical properties of saline-alkali soil and the microbial community, we constructed relevant cluster heatmaps (Figure 6). AK, AN, AP, and OM were positively correlated with *Bacillota*, *candidate division* CPR2, *Uroviricota, Blastocladiomycota*, and *Preplasmiviricota*, while negatively correlated with *Candidatus Firestonebacteria*, *Candidatus Shapirobacteria*, *Ascomycota*, *candidate division* WOR-3, and *Candidatus Beckwithbacteria*. In addition, pH was positively correlated with *Candidatus Firestonebacteria*.

Furthermore, the redundancy analysis (RDA) indicates that the environmental factors AK, AN, AP, and OM exhibit high correlations, and they are positively correlated with the changes in community composition of the BH-8 group and the composite microbial agent group (Figure 7). In contrast, pH values show a negative correlation with the changes in community composition of these two groups. These results are consistent with our analysis of soil physicochemical properties (Figure 2 and Appendix A).

### 3.7. Effects on Gene Functions of Soil Microbial Communities

The number of genes encoding carbohydrate-active enzymes (CAZymes) of different functional classes is shown in Figure 8a. Notably, the highest number of genes encode glycosyl transferases (GTs), which play a crucial role in glycosylation modifications. This is followed by genes encoding glycoside hydrolases (GHs), typically involved in the degradation of cellulose and other carbohydrates, and carbohydrate-binding modules (CBMs), which enhance the efficiency of substrate recognition and enzyme binding. This indicates the important role of GT, GH, and CBM in carbohydrate metabolism.

To further understand the effect of the composite microbial agent on soil microbial community function, we analyzed the expression differences of enzyme families in different samples. The metadata analysis revealed differentially expressed classes such as GH3, GH6, GH15, GH23, GH28, GH34, GH73, GT1, GT4, GT51, CBM2, CBM48, CBM50, and CBM91 (Figure 8b and Appendix A). Their detailed information is referred to in Appendix A. After the treatment of strain BH-8 and the composite microbial agent, the expression of genes encoding GH3, CBM50, GH23, GT51, and GH73 was found to be increased, while the expression of genes encoding GT4, GT1, CBM2, GH15, CBM91, GH34, GH6, CBM48, and GH28 was found to be decreased. Particularly, the application of the composite microbial agent had a more significant impact on the gene functions within the soil microbial community (Figure 8b). In addition, the histogram of relative functional abundance showed the distribution of these differentially expressed classes in different groups (Appendix A). These findings indicate that the application of strain BH-8 may affect the expression of many gene-encoding enzymes involved in the synthesis, degradation, and modification of carbohydrate within the microbial community. Such changes in gene expression can subsequently alter microbial metabolic pathways, potentially leading to rehabilitating soil characteristics in saline-alkali environments.

## 4. Discussion

As a promising microbial resource, *Bacillus halophilus* has demonstrated unique adaptability to high-salt environments in terms of the stability of the structural and functional components of the cell membrane and cell wall, the enzymatic properties of the reaction kinetics, the metabolic pathways, and the protein and nucleic acid components and configurations of signal transduction [38,39]. In this paper, we use the *Bacillus halophilus* BH-8 isolated from coastal saline alkaline soil combined with coal gangue to prepare a composite microbial agent. After the application of the composite microbial agent, the increased contents of available nitrogen, available phosphorus, available phosphorus and organic matter in saline-alkali soils probably improve soil fertility and ecological function. Previous report indicated that soil nitrogen (N), phosphorus (P), and potassium (K) are important constant nutrients that can limit or jointly limit plant growth [40]. Plants need nitrogen elements to support protein synthesis, chlorophyll formation, and other processes, all of which are crucial for their normal growth and development [41]. Large amounts of available phosphorus can help maintain plant growth and metabolism, with soil-available phosphorus being a limiting nutrient in many natural ecosystems [42]. Potassium has an important impact on plant growth and development, and its increased content helps to enhance plant stress tolerance and yield [43]. Furthermore, soil organic matter plays an important role in soil fertility and structure, and it is one of the main components of the global carbon cycle, playing a significant role in gas exchange [44]. Our results also show that the composite microbial agent significantly reduced soil pH in the first 7 days, which helped to improve the alkaline environment, but the effect decreased after 14 days, which may be related to the salt loss caused by irrigation [45]. In addition, the application of the composite microbial agent can effectively improve the enzyme activity in saline-alkali soils. Enzymes of various types in the soil play an important role in the normal growth of soil microbial communities and in nutrient metabolism; for example, urease activity is closely related to the nitrogen cycle in the soil [46]; phosphatase is an extracellular enzyme that can protect them from degradation [47]; catalase can promote the decomposition of hydrogen peroxide into water and oxygen, which is an important system in the soil for synthesizing and preventing the toxicity of hydrogen peroxide to soil enzymes [48]. Therefore, an improvement in enzyme activities such as phosphatase and urease in the soil can help promote the soil nitrogen cycle, enhance the protection of soil microbial bodies, and play a role in reducing soil toxicity.

The soil microbial community plays an important role in nutrient circulation and overall soil condition, and variations in soil nutrient concentration are strongly related to the soil microbial community structure [49,50]. Many previous studies have shown that coal gangue can alter soil properties and influence the bacterial and fungal community structure in the soil [51]. Our research indicates that using coal gangue alone has a limited effect on the microbial community structure of saline-alkali soils. This could be due to the physicochemical features of coal gangue, or the fact that the microbial community did not have enough time to respond to the addition of coal gangue. We found that combining coal gangue with *Bacillus halophilus* BH-8 significantly alters the microbial community structure of saline-alkali soils. Particularly, there was a significant increase in the abundance of *Bacteroidota*. *Bacteroidota* is an important indicator of soil quality, and its abundance is directly related to soil fertility [52]. Furthermore, previous studies have also highlighted the key role of *Bacteroidota* in the nitrogen cycle of soil [53]. Therefore, the increased soil fertility following the treatment of the composite microbial agent might be attributed to the relative abundance of *Bacteroidota* in the soil. *Pseudomonadota* has disease prevention and growth promotion effects on various plants [54], enhancing plant stress resistance [55], being environmentally friendly, and non-pathogenic to organisms [56]. BH-8 belongs to the genus *Bacillota*, in which the *Bacillus subtilis* has been proven to improve the bacterial community and enhance the absorption of crop nutrients [57]. *Uroviricota* includes most of the soil bacteriophages, which are very important for the soil as they can promote the stability of bacterial communities and provide more diverse auxiliary genomic functions for bacteria and plant communities [58]. However, the soil microbial community structure also varies with climate, crop types, and fertilizers [59,60]. Consequently, the impact of a composite microbial agent on the dynamics of soil microbial communities may differ depending on soil conditions and agricultural practices. Nevertheless, it is important to note that maintaining soil pH closer to neutrality, as contributed by our novel composite microbial agent, is generally universally beneficial for most crops.

Previous studies have shown that coal gangue or biological agents have a certain degree of effect on the soil microbial community function [61,62]. In this study, the CAZy database annotation revealed the application of BH-8 and the composite microbial agent resulted in changes in the gene composition encoding carbohydrate-active enzymes. Owing to its unique molecular mechanisms, *Bacteroidota* possesses the remarkable ability to secrete a diverse array of carbohydrate-active enzymes that target the highly variable glycans present in soil [63]. Microbial CAZymes play a significant role in the degradation of soil organic matter [64]. Consequently, the application of strain BH-8 and the composite microbial agent may alter the structure of microbial communities, increase the abundance of microorganisms such as *Bacteroidota*, and modify the expression levels of CAZymes, thereby enhancing the ability to decompose organic matter and improving soil fertility.

Notably, applying strain BH-8 in conjunction with coal gangue produces a synergistic effect. This phenomenon is probably due to the physicochemical properties of coal gangue. SEM analysis also confirmed that large particle size gangue favors the attachment of bacteria (Appendix A), which may create a potentially more favorable microenvironment for microbial colonization under saline-alkali stress. Such alterations may promote better functioning of BH-8, thereby improving soil nutrient cycling and physicochemical properties. However, the generalizability of these effects to diverse ecosystems and their long-term stability require further validation through multi-seasonal field trials.

## 5. Conclusions

In this study, we successfully improved the acid-base balance and nutrient environment of saline-alkali soils by using *Bacillus halophilus* BH-8 and coal gangue, which had a profound impact on the composition and gene function of the soil microbial community, effectively enhancing the fertility of saline-alkali soils and demonstrating the application value of halophilic *Bacillus* and coal gangue in the biological remediation of saline-alkali soils. These results confirm the potential of this novel composite microbial agent for soil rehabilitation, providing possible strategies for promoting plant growth in saline-alkali lands. However, the mechanisms underlying the saline-alkali soil-improving effects of BH-8, as well as its interactions with the soil microbial community, still require further investigation. Moreover, the long-term effects of this composite microbial agent on saline-alkali soils also need further exploration.

## Figures and Tables

**Figure 1 microorganisms-13-00532-f001:**
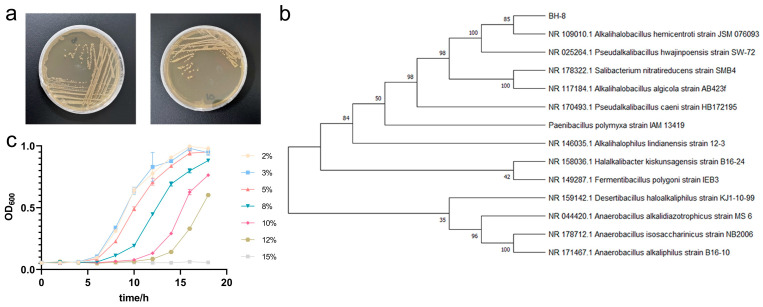
Identification and characteristics of *Bacillus halophilus* BH-8. (**a**) Colony morphology of BH-8. (**b**) Phylogenetic tree of BH-8 strain based on 16S rDNA analysis system. (**c**) Growth of BH-8 at indicated NaCl concentration.

**Figure 2 microorganisms-13-00532-f002:**
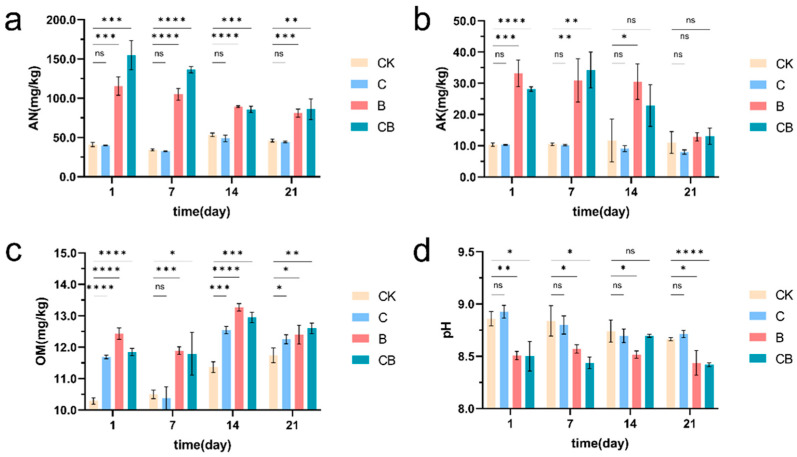
Effects on physicochemical properties of soil. (**a**) Changes in soil available nitrogen (AN) content. (**b**) Changes in soil available potassium (AK) content. (**c**) Changes in soil organic matter (OM) content. (**d**) Changes in soil pH. “CK” represents the control group without treatment. “C” represents the group treated with coal gangue, “B” represents the group treated with the BH-8 microbial agent, and “CB” represents the group treated with the composite microbial agent. * Represents *p* < 0.05, ** represents *p* < 0.01, *** represents *p* < 0.001, and **** represents *p* < 0.0001, “ns” represents "no significance".

**Figure 3 microorganisms-13-00532-f003:**
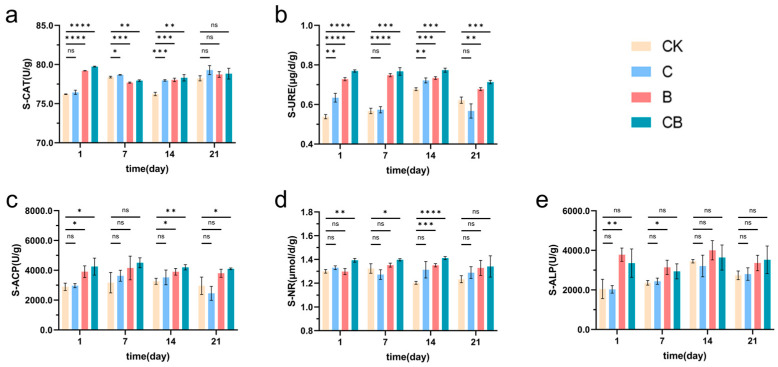
Effects on soil enzyme activity. (**a**) Changes in soil catalase activity. (**b**) Changes in soil urease activity. (**c**) Changes in soil acid phosphatase activity. (**d**) Changes in soil nitrate reductase activity. (**e**) Changes in soil alkaline phosphatase activity. “CK” represents the control group without treatment. “C” represents the group treated with coal gangue, “B” represents the group treated with the BH-8 microbial agent, and “CB” represents the group treated with the composite microbial agent. * Represents *p* < 0.05, ** represents *p* < 0.01, *** represents *p* < 0.001, and **** represents *p* < 0.0001, “ns” represents “no significance”.

**Figure 4 microorganisms-13-00532-f004:**
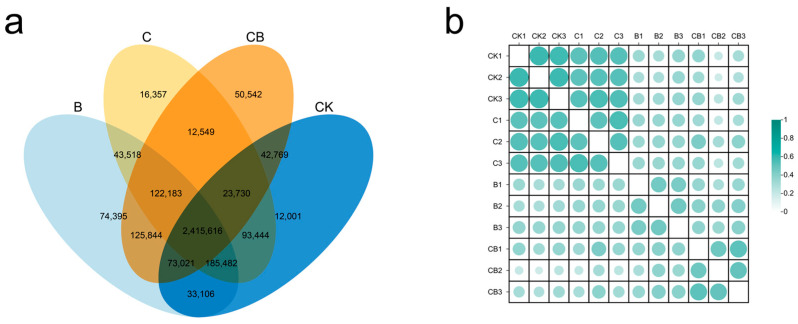
Changes in bacterial community structure and diversity in soil. (**a**) Wayne diagram represents the number of shared and unique genes among different samples. (**b**) Heat map of correlation coefficients between different samples. “CK” represents the control group without treatment. “C” represents the group treated with coal gangue. “B” represents the group treated with the BH-8 microbial agent. “CB” represents the group treated with the composite microbial agent.

**Figure 5 microorganisms-13-00532-f005:**
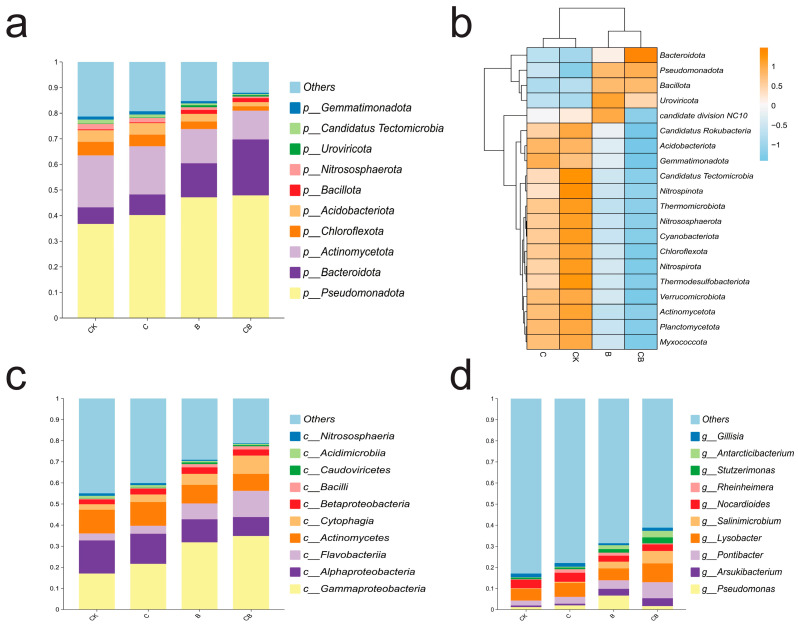
Effects on soil microbial community composition. (**a**) Histogram of relative species abundance at the phylum level. (**b**) The heatmap showing the abundance of the top 20 phyla. (**c**) Histogram of relative species abundance at the class level. (**d**) Histogram of relative species abundance at the genus level. “CK” represents the control group without treatment. “C” represents the group treated with coal gangue. “B” represents the group treated with the BH-8 microbial agent. “CB” represents the group treated with the composite microbial agent.

**Figure 6 microorganisms-13-00532-f006:**
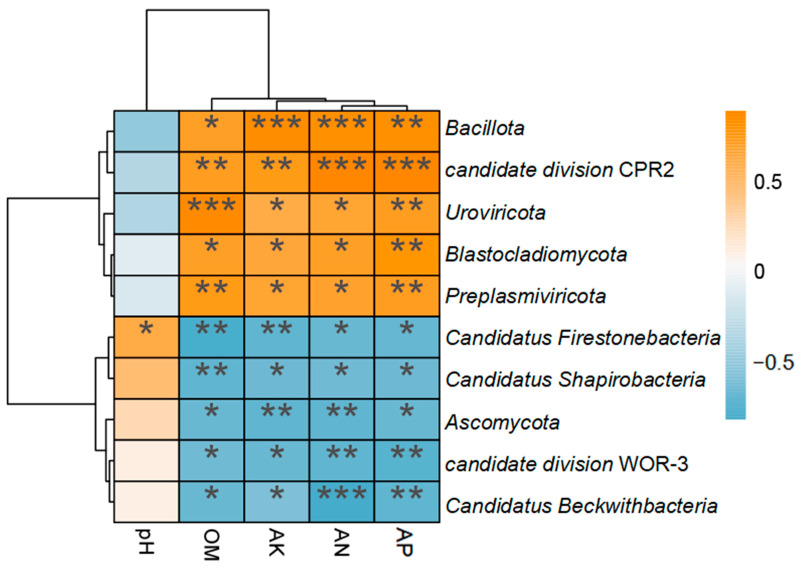
Soil physicochemical properties and species correlation heatmap. pH, OM, AK, AN, and AP respectively represent soil pH, organic matter, available potassium, available nitrogen, and available phosphorus. * Represents *p* < 0.05, ** represents *p* < 0.01, and *** represents *p* < 0.001.

**Figure 7 microorganisms-13-00532-f007:**
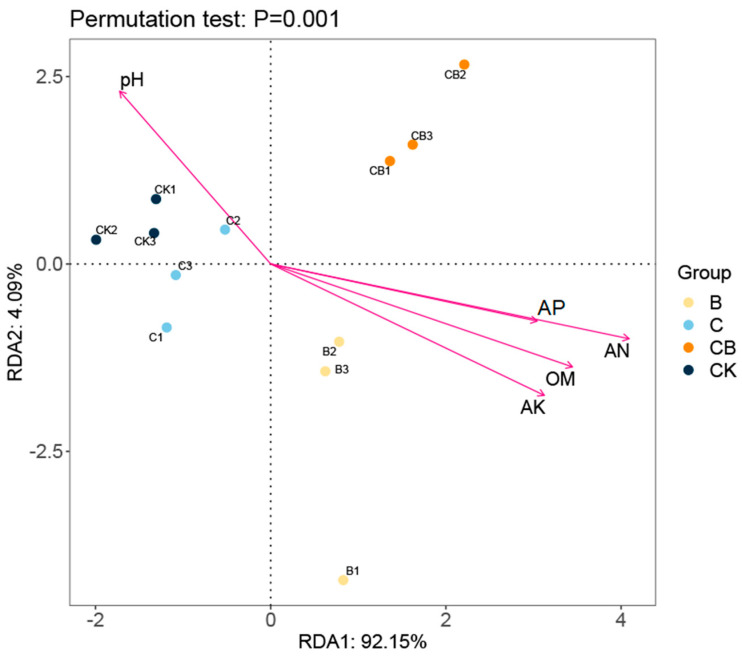
Redundancy analysis (RDA) based on the phylum level. “CK” represents the control group without treatment, “C” represents the group treated with coal gangue, “B” represents the group treated with the BH-8 microbial agent, and “CB” represents the group treated with the composite microbial agent. Each group includes three independent replicates.

**Figure 8 microorganisms-13-00532-f008:**
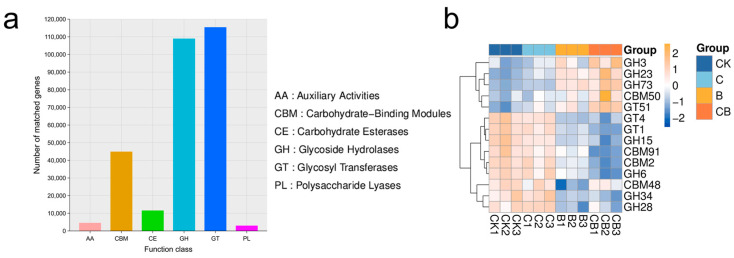
Effects on gene function of soil microbial communities. (**a**) CAZy Database annotation Gene Number. (**b**) Functional abundance clustering heat map. “CK” represents the control group without treatment. “C” represents the group treated with coal gangue. “B” represents the group treated with the BH-8 microbial agent. “CB” represents the group treated with the composite microbial agent.

**Table 1 microorganisms-13-00532-t001:** Differences in alpha diversity index between groups.

	Shannon	Simpson	Chao1	ACE
CK	6.785 ± 0.183	0.962 ± 0.005	3264.128 ± 19.229	3246.862 ± 16.566
C	6.469 ± 0.132	0.955 ± 0.006	3344.226 ± 15.617	3323.715 ± 12.102
B	5.983 ± 0.201	0.950 ± 0.007	3497.598 ± 6.757	3477.981 ± 5.112
CB	5.426 ± 0.148	0.933 ± 0.005	3451.400 ± 43.283	3429.857 ± 40.909

## Data Availability

The original contributions presented in this study are included in the article/Appendix A. Further inquiries can be directed to the corresponding authors.

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
