# Peer review of "Bacillus halophilus BH-8 Combined with Coal Gangue as a Composite Microbial Agent for the Rehabilitation of Saline-Alkali Land"

_microorganisms, 2025, doi:10.3390/microorganisms13030532_

Round 1
Reviewer 1 Report
Comments and Suggestions for Authors
In this work, a technique was developed to improve the pH values ​​and growth conditions of crops planted in alkaline soils. The results are promising and its implementation in other parts of the planet with a similar problem would require additional studies. The work is well planned and articulated.
However, there are some considerations that authors must address.
It is not mentioned in the document but assuming it is a species of Bacillus, is it a spore producer?
The meaning of the abbreviation w/t must be specified in the document.
In the text it says
“In addition, smaller grain sizes of coal gangue resulted in higher growth inhibition of BH-8 (Fig- 217
ure 2b).”
This is an interesting result assuming that the area/volume ratio under these conditions is the greatest of all, from which it could be inferred that the diffusion of elements and salts would have greater exposure to the culture, and therefore would exert a greater toxic effect, being able to access the cytoplasm. cellular and affecting the reproduction process.
Regarding larger particle sizes, were microscopy photos taken in order to find out if these small solids acted as a support for the generation of microcolonies?
A pertinent question in this line of reasoning refers to whether the cells grow isolated or loose in the culture medium or flocculate generating small clumps in the culture, or are associated with the small solids added in the coal gangue. This type of idea should in my opinion be included in the discussion.
In figure two, the symbols representing the different conditions are too small, so that one condition is not effectively distinguished from the other.
A relevant analysis that I understand would be convenient to do refers to using sterile soil, without any additional microbial component to evaluate the effect on the different soil conditions and pH, since there are technologies that could be used to sterilize the soil and repopulate with populations. of this bacteria
In the text it says “The soil microbial community plays an important role in nutrient circulation and overall soil condition, and variations in soil nutrient concentration are strongly related to soil microbial community structure
In this sense, it is pertinent to indicate that the composition and diversity of the species in the crop fields depends on the climatic conditions, and the type of crop that is intended to be produced, as well as the addition of fertilizers, which will affect the composition of the microbial community and therefore the expected effect on crop growth and yield is very unpredictable. I understand that focusing on maintaining pH values ​​far from a very alkaline value will generally have a positive effect on plant growth and therefore on crop yields. This type of idea should in my opinion be included in the discussion.
In the text it says
“This is likely because coal gangue offers a more suitable habitat for microorganisms, promotes changes in microbial communities structure, thereby improving the soil structure and its physiological and physicochemical properties.”
Talking about this technique promoting a more suitable habitat, I understand that it could only be stated under a comparative context using different crops and different climates, the concept " suitable " I understand that lacks scientific rigor. The change that promotes the presence of bacterial groups that improve yields depends on their being present in the ecosystems and this type of statement in any case requires an analysis over longer periods of time and under seasonal, tropical, etc. climatic conditions. This type of idea should in my opinion be included in the discussion
Author Response
We thank you for your constructive comments and appreciation for the importance of our study. We have carefully revised the manuscript and uploaded a “Revised Manuscript” file (changes in the manuscript are marked in red), a comparison copy of the manuscript as a "Marked-Up Manuscript" file, and a “Revised Supplemental Material” file. Here are the point-by-point responses:
- It is not mentioned in the document but assuming it is a species of Bacillus, is it a spore producer?
Reply: Thank you for your precious suggestion. We have conducted microscopic observations to confirm the spore-forming capability of BH-8. Spore-like structures were observed in cultures of BH-8 under the microscope with spore staining, which confirmed that BH-8 is a spore producer. This characteristic probably enhances its survivability in harsh saline-alkali environments. The microscopic picture was added in the revised supplementary material (Revised Supplementary Material, Figure S1). We also supplemented the description of experimental methods in section 2.5 (Revised Manuscript, lines 154-158), and added the analysis of results in the revised manuscript (Revised Manuscript, lines 225-227):
“Microscopy analysis confirmed that BH-8 is a spore-forming bacterium (Figure S1). This characteristic probably enhances its survivability in harsh saline-alkali environments.”
- The meaning of the abbreviation w/t must be specified in the document.
Reply: We sincerely apologize for the oversight. This abbreviation has now been clarified in Section 2.4 (Revised Manuscript, lines 151-152):
“The wt% refers to the weight percentage of coal gangue in the culture.”
In addition, we revised “w/v” to “weight / volume” (Revised Manuscript, lines 167).
- In the text it says “In addition, smaller grain sizes of coal gangue resulted in higher growth inhibition of BH-8 (Figure 2b).” This is an interesting result assuming that the area/volume ratio under these conditions is the greatest of all, from which it could be inferred that the diffusion of elements and salts would have greater exposure to the culture, and therefore would exert a greater toxic effect, being able to access the cytoplasm. cellular and affecting the reproduction process. Regarding larger particle sizes, were microscopy photos taken in order to find out if these small solids acted as a support for the generation of microcolonies? A pertinent question in this line of reasoning refers to whether the cells grow isolated or loose in the culture medium or flocculate generating small clumps in the culture, or are associated with the small solids added in the coal gangue. This type of idea should in my opinion be included in the discussion.
Reply: Thank you very much for your precious suggestion. We fully agree that the smaller grain sizes of coal gangue result in the higher area/volume ratio, from which it could be inferred that the diffusion of elements and salts would have greater exposure to the culture, and therefore would exert a greater toxic effect, being able to access the cytoplasm and affecting the reproduction process. We have added this part (Revised Manuscript, lines 239-244), and now it reads:
“The inhibition of BH-8 growth by smaller coal gangue particles can be attributed to their higher surface area to volume ratio, which enhances the dissolution and diffusion of salts and potential heavy metals into the culture medium. This probably facilitates the influx of toxic elements into the cytoplasm, and impairs metabolic processes critical for bacterial reproduction. Conversely, larger particles exhibit a lower exposure to the culture medium, thereby exerting a reduced toxic effect.”
In addition, we performed Scanning Electron Microscope (SEM) to observe bacterial colonization on coal gangue particles. The results showed that BH-8 cells were more likely to attach to larger coal gangue particles (1 mm) and form microcolonies (Figure S3), suggesting that larger gangue particles support microbial colonization. This may be because larger particles have deeper and larger pores that are favorable for bacterial colonization. We have added this part in the revised manuscript (Revised Manuscript, lines 245-250), and now it reads:
“SEM analysis further revealed that BH-8 cells preferentially adhered to larger gangue particles (1 mm) and form microcolonies (Figure S3), suggesting that larger gangue particles support microbial colonization. This may be because larger particles have deeper and larger pores that are favorable for bacterial adhesion.”
- 4. In figure two, the symbols representing the different conditions are too small, so that one condition is not effectively distinguished from the other.
Reply: We sincerely apologize for the indistinguishable symbols. We have revised this figure to enlarge the symbols and improve color contrast for clarity (Revised Supplementary Material, Figure S2).
- A relevant analysis that I understand would be convenient to do refers to using sterile soil, without any additional microbial component to evaluate the effect on the different soil conditions and pH, since there are technologies that could be used to sterilize the soil and repopulate with populations.
Reply: Thank you very much for your valuable suggestions. Your insightful recommendation will undoubtedly strengthen our understanding of the BH-8’s function and refine its application strategies. We fully agree that experiments using sterilized soil could further identify the specific contributions of BH-8. In this study, we primarily focused on natural saline-alkali soil conditions (non-sterile) to develop a composite microbial agent for the rehabilitation of saline-alkali land. However, to better understand the critical role of the BH-8 strain, we will assess the performance of BH-8 in more controlled environments, such as sterile soil, in future studies.
- In the text it says “The soil microbial community plays an important role in nutrient circulation and overall soil condition, and variations in soil nutrient concentration are strongly related to soil microbial community structure.”. In this sense, it is pertinent to indicate that the composition and diversity of the species in the crop fields depends on the climatic conditions, and the type of crop that is intended to be produced, as well as the addition of fertilizers, which will affect the composition of the microbial community and therefore the expected effect on crop growth and yield is very unpredictable. I understand that focusing on maintaining pH values ​​far from a very alkaline value will generally have a positive effect on plant growth and therefore on crop yields. This type of idea should in my opinion be included in the discussion.
Reply: Thank you very much for highlighting this critical point. We have added this part with the corresponding references in the discussion (Revised Manuscript, lines 498-503):
“However, the soil microbial community structure also varies with climate, crop types, and fertilizers [59, 60]. Consequently, the impact of a composite microbial agent on the dynamics of soil microbial communities may differ depending on soil conditions and agricultural practices. Nevertheless, it is important to note that maintaining soil pH closer to neutrality, as contributed by our novel composite microbial agent, is generally universally beneficial for most crops.”
- In the text it says“This is likely because coal gangue offers a more suitable habitat for microorganisms, promotes changes in microbial communities structure, thereby improving the soil structure and its physiological and physicochemical properties.”Talking about this technique promoting a more suitable habitat, I understand that it could only be stated under a comparative context using different crops and different climates, the concept " suitable " I understand that lacks scientific rigor. The change that promotes the presence of bacterial groups that improve yields depends on their being present in the ecosystems and this type of statement in any case requires an analysis over longer periods of time and under seasonal, tropical, etc. climatic conditions. This type of idea should in my opinion be included in the discussion.
Reply: Thank you for your precious suggestion. We sincerely apologize for the inappropriate descriptions and we fully agree that the term "suitable" requires scientific validation across broader environmental variables, such as crop types, climatic conditions, and long-term field studies. We have revised this part in the discussion in the revised manuscript (Revised Manuscript, lines 515-522), and now it reads:
“Notably, applying strain BH-8 in conjunction with coal gangue produces a synergistic effect. This phenomenon probably due to the physicochemical properties of coal gangue. SEM analysis also confirmed that large particle size gangue favors the attachment of bacterial (Figure S3), which may create a potentially more favorable microenvironment for microbial colonization under saline-alkali stress. Such alterations may promote better functioning of BH-8, thereby improving soil nutrient cycling and physicochemical properties. However, the generalizability of these effects to diverse eco-systems and their long-term stability requires further validation through multi-seasonal field trials.”

Reviewer 2 Report
Comments and Suggestions for Authors
The manuscript describes research on the usefulness of coal gangue and halophilic bacteria isolated from saline soil for the remediation of saline-alkaline soils. This topic fits into the scope of the journal Microorganisms. The authors used the correct methods and obtained very interesting results, which they correctly developed statistically and presented in numerous figures and tables. They correctly discussed the results in the light of the latest literature in this field. Therefore, I recommend the work for publication in the journal Microorganisms after minor corrections. Minor comments are provided below.
Introduction
Lines 100-102: This statement should be included in the conclusions or possibly in the discussion, but not here. I recommend changing the sentence to a research hypothesis and then leaving it here.
Materials and Methods
Lines: 148-149; 158-160; 165-167; 175-177 - move this information to the subsection "Statistical analysis"
Line: 162 - no reference to the method and a brief description of the method
Lines 163-165 no description of the method explain what it is
Results
lines 208-221 should be included in Materials and methods in subsection 2.4
Figure 2. - move to Supplemental material
Figure 3. missing explanations of abbreviations AN, AK, OM
Figure 4. why was alkaline phosphatase activity not included, I recommend moving it from supplemental material
Figure 7. explanations of abbreviations CK, C, CB, B unnecessary
Figure 8. missing explanations of abbreviations c1, c2 etc.
Discussion
The authors obtained very valuable and interesting results, therefore I recommend underlining them in the Conclusions section. For example, paragraph lines 494-502 and the last sentence of the introduction should be reworded into a Conclusions chapter.
Author Response
We thank you for your constructive comments and appreciation for the importance of our study. We have carefully revised the manuscript and uploaded a “Revised Manuscript” file (changes in the manuscript are marked in red), a comparison copy of the manuscript as a "Marked-Up Manuscript" file, and a “Revised Supplemental Material” file. Here are the point-by-point responses:
- Lines 100-102: This statement should be included in the conclusions or possibly in the discussion, but not here. I recommend changing the sentence to a research hypothesis and then leaving it here.
Reply: Thank you for your precious suggestion. We have revised this sentence in the revised manuscript (Revised Manuscript, lines 103-105), and now it reads:
“The above analyses were used to confirm whether this novel composite microbial agent plays a role in rehabilitating saline-alkali land.”
In addition, we have added this statement in the “conclusion” section (Revised Manuscript, lines 529-531):
“These results confirmed the potential of this novel composite microbial agent for soil rehabilitation, providing possible strategies for promoting plant growth in saline-alkali lands.”
- Lines: 148-149; 158-160; 165-167; 175-177 - move this information to the subsection "Statistical analysis".
Reply: Thank you for your precious suggestion. The mentioned lines have been moved to the "Statistical analyses" subsection in the revised manuscript (Revised Manuscript, lines 195-201).
- Line: 162 - no reference to the method and a brief description of the method.
Reply: Thank you for your valuable comments and suggestions. We sincerely apologize for the oversight in not providing a brief description of it in our manuscript. We have added this part in the revised manuscript (Revised Manuscript, lines 175-178).
- 4. Lines 163-165 no description of the method explain what it is.
Reply: Thank you for your precious suggestion. We sincerely apologize for the oversight in not providing a description of these methods. We have added this part in the revised manuscript (Revised Manuscript, lines 180-185).
- lines 208-221 should be included in Materials and methods in subsection 2.4.
Reply: Thank you very much for your valuable suggestions. We have revised this part in the revised manuscript (Revised Manuscript, lines 148-152, line 234-236).
- Figure 2. - move to Supplemental material.
Reply: Thank you for your precious suggestion. Figure 2 (The growth of BH-8 co-cultured with coal gangue at varied concentrations and particle sizes) has been moved to Supplementary Figure S2 (Revised Supplementary Material, Figure S2).
- Figure 3. missing explanations of abbreviations AN, AK, OM.
Reply: We sincerely apologize for the missing explanations of abbreviations AN, AK, OM in Figure 3 (now Figure 2). We have added the definition of the abbreviation AN, AK, and OM in the figure caption (Revised Manuscript, lines 277-279).
- Figure 4. why was alkaline phosphatase activity not included, I recommend moving it from supplemental material.
Reply: Thank you for your precious suggestion. Alkaline phosphatase (ALP) activity data (previously in Supplementary Figure S2) has been incorporated into Figure 3 (previously Figure 4) in the revised manuscript.
- Figure 7. explanations of abbreviations CK, C, CB, B unnecessary.
Reply: Thank you for your precious suggestion. We have deleted the explanations of abbreviations CK, C, CB, and B in Figure 6 (previously Figure 7) in the revised manuscript.
- Figure 8. missing explanations of abbreviations c1, c2 etc.
Reply: We sincerely apologize for the missing explanations of abbreviations c1, c2 etc. In this study, “CK” represents the control group without treatment, with three replicates labeled as CK1, CK2, and CK3. “C” represents the group treated with coal gangue, with three replicates labeled as C1, C2, and C3. “B” represents the group treated with the BH-8 microbial agent, with three replicates labeled as B1, B2, and B3.“CB” represents the group treated with the composite microbial agent, with three replicates labeled as CB1, CB2, and CB3. We have revised the figure caption in the revised manuscript (Revised Manuscript, lines 412).
- The authors obtained very valuable and interesting results, therefore I recommend underlining them in the Conclusions section. For example, paragraph lines 494-502 and the last sentence of the introduction should be reworded into a Conclusions chapter.
Reply: Thank you very much for your precious suggestion. We have added a "Conclusions" section to emphasize our findings in the revised manuscript (Revised Manuscript, lines 523-534), and now it reads:
“In this study, we successfully improved the acid-base balance and nutrient environment of saline-alkali soils by using Bacillus halophilus BH-8 and coal gangue, which had a profound impact on the composition and gene function of the soil microbial community, effectively enhancing the fertility of saline-alkali soils and demonstrating the application value of halophilic Bacillus and coal gangue in the biological remediation of saline-alkali soils. These results confirmed the potential of this novel composite microbial agent for soil rehabilitation, providing possible strategies for promoting plant growth in saline-alkali lands. However, the mechanisms underlying the saline-alkali soil-improving effects of BH-8, as well as its interactions with the soil microbial community, still require further investigation. Moreover, the long-term effects of this composite microbial agent on saline-alkali soils also need further exploration.”
